# Effect of a High-Intensity Circuit Training Program on the Physical Fitness of Wildland Firefighters

**DOI:** 10.3390/ijerph20032073

**Published:** 2023-01-23

**Authors:** Jorge Gutiérrez-Arroyo, Fabio García-Heras, Belén Carballo-Leyenda, José G. Villa-Vicente, Juan Rodríguez-Medina, Jose A. Rodríguez-Marroyo

**Affiliations:** VALFIS Research Group, Institute of Biomedicine (IBIOMED), University of León, 24071 León, Spain

**Keywords:** performance, exercise, physical activity, strength training, endurance, occupational health

## Abstract

Wildland firefighting implies high physical and psychological demands for the personnel involved. Therefore, good physical fitness can help increase the work efficiency of wildland firefighters (WFFs) and safeguard their health. High-intensity circuit training (HICT) could be a good alternative to improve the physical condition of WFFs since it stands out for its functionality and economy. Therefore, the objective of this study was to analyze the effects of HICT on the WFFs’ physical fitness. The study involved 9 WFFs (8 males and 1 female; 29.8 ± 2.8 years; 175.6 ± 6.7 cm) who completed a training program and 9 WFF candidates (8 males and 1 female; 24.7 ± 6.6 yr, 176.5 ± 7.0 cm) as a control group. WFFs performed an 8-week HICT program (two weekly training sessions). The training sessions lasted approximately 45 min and were performed at an intensity >80% of maximal heart rate and RPE values >7.5. At the beginning and the end of the intervention, subjects’ physical fitness (i.e., aerobic capacity, cardiorespiratory endurance, muscular resistance and explosive strength) was assessed through different tests. After completing the training program, WFFs had significantly increased (*p* < 0.05) the speed at which the ventilatory thresholds were determined (12.4 ± 13.9 and 5.7 ± 7.3% for the ventilatory and respiratory compensation thresholds, respectively) and also their abdominal (31.2 ± 17.2%), lumbar (34.1 ± 13.5%) and upper limb (13.3 ± 16.0%) strength. Moreover, improvements (*p* < 0.05) in the explosive strength of legs (7.1 ± 5.8%) and performance in a specific physical employment test (12.2 ± 6.1%) were observed. In conclusion, the results of this study show that an 8-week high-intensity circuit training program could be an effective and safe method to improve WFFs’ physical fitness and performance.

## 1. Introduction

Wildland firefighting is highly demanding, both physically and psychologically. Several factors, including the long duration of work, orography and conditions of the deployments sites, environmental conditions, wearing personal protective equipment and lack of sleep, determine the workload of wildland firefighters (WFFs) [1,2,3,4,5] and their health and safety [6]. In addition, wildland fire suppression involves using hand tools such as building fire lines, brush removal, setting backfires and mopping up [7]. All of the above help us understand why wildland firefighting has been considered a highly demanding occupation [8], one which involves work energy consumption reaching 2628 ± 714 kcal·day^−1^ [9]. Considering the intense and diverse nature of their work, it is widely recognized that WFFs must maintain a level of physical fitness necessary to perform their work safely and efficiently [10], helping safeguard their health throughout their working life [5,11,12]. Therefore, the use of physical tests for selecting these workers can help ensure that they possess the physical fitness levels necessary to perform their work safely and efficiently [10]. Specifically, for WFFs, to determine whether personnel are fit for duty, an increasing number of agencies (e.g., the USDA Forest Service; Australian Fire Agencies; British Columbia Forest Service in Canada or Ministry of Agriculture, Fisheries and Food in Spain) employ physical competency tests such as the Pack test [13,14]. This test involves a 4.8 km hike over level terrain while carrying a 20.4 kg pack within 45 min. This test was designed to challenge an individual’s muscular strength and cardiorespiratory fitness, mimicking the physiological strain encountered during wildland fire suppression using hand tools [13].

The inclusion of training programs in physically demanding occupations might be designed to improve not only physical fitness but also job performance, as previously reported in the military [15,16,17,18], structural firefighting [19,20,21,22,23,24] and specifically for WFFs [11,25,26,27,28]. Although training programs have traditionally focused on increasing the cardiovascular fitness of workers, the importance of selecting adequate intensities to improve both aerobic and anaerobic fitness [19], as well as muscular strength and endurance, has been highlighted [24]. In recent years, high-intensity training has been gaining popularity in physically demanding occupations [29,30]. This type of training involves repeated bouts of high-intensity effort (i.e., 85–95% of the maximal heart rate) followed by varied recovery times with a session duration of ~40 min [31]. Results in the literature demonstrate that high-intensity training increases both aerobic and anaerobic capacity [32,33], as well as muscular power [34,35,36], with less training volume or time required to achieve greater cardiovascular and muscular adaptations [37]. These characteristics make high-intensity training a time-effective method to match the limited and unpredictable time schedules of firefighters in the workplace [38]. In this sense, high-intensity training for firefighters has been proven to increase metabolic rate, cardiac output and aerobic capacity, along with greater adherence compared to a regular training program [39,40]. In the military, the implementation of high-intensity training has shown improvements in physical capacity and muscular and metabolic condition, with a substantial improvement in the coping ability of the specific physical demands inherent to their work activity [16]. 

Despite these benefits, traditional high-intensity protocols are limited regarding specific preparation for the typical tasks of firefighters on duty [24,41,42]. In recent years, high-intensity functional training has emerged as a variation of traditional high-intensity training. This approach relies on multiple and multimodal functional exercises such as lifting, pushing, pulling, loading or locomotion that more closely resemble work-specific tasks [16]. One form of high-intensity functional training is high-intensity circuit training (HICT), where participants typically complete a set of exercises for a certain number of repetitions or time, each of which targets a different muscle group following a circular fashion [43]. This type of training has been previously applied in structural firefighters through multi-joint exercises that simulate the movement patterns of firefighting tasks in circuit mode [19]. Abel et al. [19] compared the achieved aerobic and anaerobic intensities of the HICT workout to physiological data previously reported on firefighters performing fire suppression and rescue tasks. Results showed that the circuit-based workout produced lower cardiovascular stress but similar anaerobic stress compared to firefighting tasks. Recently, Chizewski et al. [22] found significant improvements in both fitness and firefighter ability in recruit firefighters following a 7-week daily routine of high-intensity functional training (60 min, 5 days·wk^−1^). While some research has compared the effectiveness of HICT versus more traditional unimodal training in military populations [17], no research has been conducted to examine the impact of HICT on WFFs. Therefore, this study aimed to determine the effect of an HICT program on physical fitness and job performance in a group of Spanish WFFs.

## 2. Materials and Methods

### 2.1. Participants

The study involved a sample of 9 WFFs (8 males and 1 female; 29.8 ± 2.8 yr, 175.6 ± 6.7 cm) belonging to a Spanish crew in northwestern Spain and 9 WFF candidates (8 males and 1 female; 24.7 ± 6.6 yr, 176.5 ± 7.0 cm). Participants were healthy and physically active (e.g., endurance exercise 45–60 min per training session three times per week) and all had prior experience in regular strength and endurance training programs (>1 yr). WFFs had a work experience in wildland fire suppression of 7.4 ± 4.5 years. Throughout the research period, subjects were encouraged to maintain their dietary patterns and physical activity routines. Written informed consent was obtained from the participants before starting the study. The test protocol was developed according to the Declaration of Helsinki guidelines for research on human subjects and was approved by the Ethics Committee of the University of León (025-2020, 22 July 2020).

### 2.2. Experimental Design

Using a nonrandomized study design, subjects were divided into two training groups according to the training program performed during an intervention period of 8 weeks. Thus, WFFs followed a HICT program (Table 1), while WFF candidates were assigned to the control group (CG). For the duration of the study, the CG was enrolled in a specific training camp where they were trained in the work techniques commonly used by WFFs. During this period, they performed low-to-moderate intensity physical activity (i.e., rating of perceived exertion (RPE) <6 using a CR 0–10 scale) on a regular basis for 2 h·wk^−1^. Prior to the intervention, all participants underwent a familiarization session with the test’s protocols used in the study. In addition, WFFs were familiarized with the training program exercises. Subjects’ physical fitness was assessed, by the same researchers, one week before starting the study and at the end of the intervention period. The tests were performed over three testing sessions interspersed by 24 h in between. In the first laboratory session, subjects’ anthropometric, aerobic capacity and cardiorespiratory endurance were assessed. In the second session, participants performed a battery of tests to determine the strength of different muscle groups. Finally, during the last session, subjects performed the Pack test simultaneously. All testing sessions were performed at the same time of day under similar environmental conditions. All tests were preceded by a standardized 15 min warm-up period of submaximal running and free stretching. Subjects were not allowed to consume products containing caffeine during the preceding 2 h. In the 24 h before testing sessions, subjects were instructed to avoid strenuous physical activity.

#### 2.2.1. High-Intensity Circuit Training Program

WFFs trained 2 days per week during the 8-week training intervention. Participants had to complete >90% of all training sessions to be included in the final analyses. The training sessions were approximately ~45 min and included muscular actions or specific movements performed by the WFFs during wildfires suppression. WFFs were instructed to perform the programmed exercises in each session (Table 1) at intensities >80% of maximal HR and RPE values >7.5 [31] for as many repetitions as possible. During all training sessions, HR was monitored every 5 s (Polar RS800CX, Polar Electro Oy, Kempele, Finland) and RPE was obtained immediately after the completion of each exercise, using the Borg CR 0–10 scale [44]. The work:rest ratio was modified through the intervention period. It started with a ratio for beginners (1:2), then from the second week the ratio was modified to a higher load pattern (1:1), and from the fourth week it was changed to a much more intense load ratio (2:1) [45].

The training sessions were comprised of four different parts: (i) a general warm-up phase involving joint mobility exercises (5–10 min); (ii) a specific warm-up consisting of muscle activation and cardiopulmonary activation (~10 min); (iii) a central part consisting of 30–40 min of HICT where subjects were encouraged to perform as many exercises as possible at a high intensity (>80% maximal HR). The consecutive exercises within a set involved alternating muscle groups to avoid muscle fatigue (Table 1) and to develop the physical capabilities required for wildland firefighting (e.g., aerobic capacity, muscular strength, power, flexibility and agility) [46]; and (iv) a cooldown phase consisting of ~5 min of active stretching and myofascial release with a foam roller.

#### 2.2.2. Physical Fitness Tests

The WFFs performed a graded exercise test on a treadmill (h/p/cosmos pulsar, Cosmos Sports & Medical GMBH, Nussdorf-Traunstein, Germany) to assess their VO_2max_ and determine their ventilatory thresholds. The test started at 6 km·h^−1^, with the speed increased by 1 km·h^−1^ every 1 min until volitional exhaustion. The slope of the treadmill was kept constant at 1%. Breath-by-breath gas exchange and heart rate (HR) were continuously monitored throughout the trial with a 12-lead electrocardiogram (Medisoft Ergocard, Medisoft Group, Sorinnes, Belgium). The VO_2max_ and the maximal HR were the highest values obtained during the last 30 s before exhaustion. Criteria for the determination of maximal oxygen uptake were [47]: VO_2_ plateau (≤150 mL min^−1^), RER ≥ 1.15, maximal HR of ±10 beats of maximal HR predicted for age (220–age) and RPE ≥ 8. Maximum speed was determined as the highest speed the subjects could maintain during a complete stage, plus the interpolated speed from incomplete stages [48]. Ventilatory threshold (VT) and respiratory compensation threshold (RCT) were identified according to the following criteria [49]: increase in both ventilation and oxygen equivalent (VE·VO_2_^−1^) and end-tidal oxygen without a concomitant increase in the ventilatory equivalent to carbon dioxide (VE·VCO_2_^−1^) for the VT, and an increase in both VE·VO_2_^−1^ and VE·VCO_2_^−1^ along with a decrease in the pre-end-tidal carbon dioxide emission for the RCT.

During the second assessment session, five tests were performed to assess the strength fitness of different parts of the body. The Biering–Sørensen test was used to measure the isometric resistance of the trunk extensor muscles [50,51]. This test consisted of the subject being placed on a stretcher in the prone position, aligning the iliac crests with the edge of a stretcher. The trial ends if the participant can maintain the upper body in a horizontal position. The maximum time each subject could maintain the posture was recorded, up to 240 s.

The upper body resistance was assessed using the Push-up test [52]. Subjects were placed in plank position with hands shoulder-width apart and elbows fully extended. Participants were required to complete as many push-ups as possible (e.g., chest touching the mat) until exhaustion, without rest, or until two consecutive push-ups were performed incorrectly or with inadequate technique. The total number of correctly completed push-ups was determined [12]. 

The Plank test was performed to assess the core muscles’ resistance, for which a subject has to maintain their body in a plank position above the ground, supported on their toes and forearms. Subjects’ elbows were kept shoulder-width apart with their hands clenched in fists in front of their faces. The test ended when participants could not maintain the position of the pelvis or shoulder girdle. The time they held the posture was measured up to a maximum of 240 s [12,51].

The lower extremity explosive strength was assessed using a countermovement jump [53,54]. Subjects performed the jump while keeping their hands on their hips. The jump height was determined using the validated mobile phone application My Jump [55]. This app was developed to calculate the jump height from flight time using the high-speed video recording facility on the iPhone 5s. Three jumps were performed and the best value was recorded for subsequent analysis.

Handgrip strength was measured using a dynamometer (TKK 5401, Takei Scientific Instruments Co., Ltd., Nigata, Japan). Subjects were instructed to squeeze the device as hard as possible while keeping the elbow flexed to 90° and the forearm in a neutral position while sitting [56]. Both the handgrip strength of the right and left hand were assessed.

Finally, participants performed the Pack test on an athletics track. Subjects completed 12 laps (4.8 km) carrying a 20.4 kg backpack [13,14,57]. Although in the original test the subjects must complete the distance in less than 45 min [57], they were instructed to complete the test in the shortest time without actually running. Verbal encouragement was provided to the participants throughout the test; however, no feedback about lap times was given at any stage [13]. The HR response was recorded continuously every 5 s (Polar RS800CX, Polar Electro Oy, Kempele, Finland). In addition, the RPE was obtained during the last 10 m of each lap using the Borg CR 0–10 scale [44]. The Pack test performance was measured using a photocell timing system (DSD Laser System, DSD Inc., León, Spain).

### 2.3. Statistical Analysis

The results are expressed as mean ± standard deviation (*SD*). The assumption of normality was verified using the Shapiro–Wilk test. Changes in physical fitness and the Pack test performance were examined by a repeated-measures two-way analysis of variance (time [pre-test vs. post-test] × group [HICT vs. CG]). In addition, a one-way analysis of covariance was used to establish differences between groups’ relative changes in performance, using the pre-test values as covariates. When a significant *F* value was found, Bonferroni’s post hoc test was used to establish significant differences between mean values. Values of *p* < 0.05 were considered statistically significant. The effect size was calculated using Cohen’s d test. Cohen’s d values of <0.20, 0.20–0.50, 0.51–0.80 and >0.80 were rated as trivial, small, moderate and large effects, respectively [58]. Meaningful changes (0.2 × between-subjects *SD*) were obtained to determine the effectiveness of the HICT program [59]. Limits for the true value were calculated (observed changed ± 90% confidence interval), with the intervention rated as beneficial or harmful when they lay beyond the meaningful changes [59]. Analyses were performed using SPSS+ V.25.0 statistical software (SPSS, Inc., Chicago, IL, USA).

## 3. Results

The pre-test results were similar in both groups (Table 2, 3 and 4), except for those obtained in the Sörensen and Push-up tests, whose values were significantly (*p* < 0.05) higher in GC and HICT, respectively (Table 3). However, in the assessment performed at the end of the intervention period, the percentage of VO_2max_ at which VT and RCT were determined, the percentage of maximal HR at which RCT occurred (Table 2), the Push-up (Table 3) and Pack (Table 4) tests’ performances were higher (*p* < 0.05) in HICT.

In HICT, both the speed at the VT and RCT improved significantly (*p* < 0.05) after the training program (Table 2). In the same way, the percentage of VO_2max_ and maximum HR at the RCT was higher (*p* < 0.05) in the post-test. The results obtained in the Biering–Sørensen, Push-up, Plank and CMJ tests were significantly higher (*p* < 0.05) in the post-test than in the pre-test (Table 3). Similarly, the Pack test performance was substantially improved (*p* < 0.05) after the training period (Table 4), with testing time reduced by approximately 14% (~5 min) after the intervention. On the contrary, in CG, an improvement (*p* < 0.05) in the post-test values of the Push-up and Plank test performances (Table 3) and a reduction (*p* < 0.05) in the percentage of maximal HR and speed at the VT were observed.

The changes found in the VT and RCT were greater (*p* < 0.05) in HICT than in CG (Table 2). Likewise, the change observed in the Pack test and in the muscle strength tests’ performance was significantly higher (*p* < 0.05) in HICT. Only the relative change analyzed in the Push-up test was similar between groups (~15%). The changes induced by the HICT intervention were rated as beneficial for the improvement of ventilatory thresholds, WFFs’ specific performance, Sörensen, Push-up, Plank and CMJ tests (Figure 1). The specific training camp led to beneficial changes to WFF candidates in the Push-up and Plank tests (Figure 1).

## 4. Discussion

The main finding of this study was that an 8-week HICT program applied to a group of WFFs significantly increased their physical fitness in terms of leg power, muscular resistance of arms, muscular resistance of trunk flexors and extensors, and cardiorespiratory endurance. In addition, an improvement in specific job performance was also obtained, as the completion time of the employment-competence Pack test was significantly reduced.

The results obtained agree with those previously reported, where the effectiveness of the HICT program in the improvement of muscular endurance in moderately trained populations was highlighted [17]. Heinrich et al. [17] performed a program, of the same duration as the one in this study, which was aimed at developing strength, power and speed through a circuit using resistance in the exercises (e.g., own body weight, medicine balls) employed in the military. These researchers reported an improvement of ~10% in the Push-up test after the training program, which aligns with the ~13% achieved in our study. In contrast, Chizewski et al. [22] reported substantially greater improvements (~37%) after implementing a high-intensity functional training program for firefighters. In both studies, a high-intensity training modality was used, incorporating specific movements mimicking the job tasks of firefighters and WFFs. Our training program included scheduled rests of ≤30 s between work bouts, following Klika and Jordan [60]. However, in high-intensity functional training, there were no defined rests since the goal is to perform a given number of repetitions in the shortest time possible, or a set of exercises must be completed in a determined time while performing the greatest possible number of repetitions [61]. Therefore, it is plausible that the higher number of training sessions performed in the study by Chizewski et al. [22] (35 vs. 16 training sessions) could condition the improvements obtained. Despite this circumstance, the gains obtained in abdominal strength (22%) and vertical jump (<1%) were substantially lower than those found in our work (~31 and ~7%, respectively). This fact could be mainly linked to the type of exercises applied in the training programs and, on the other hand, to the tests selected to assess abdominal resistance in both studies. While the number of sit-ups performed in 1 min was computed in the study by Chizewski et al. [22], we used the Plank test in which the subjects had to maintain their bodies in a static plank position for as long as possible. Moreover, our training program included several plyometric exercises that could have contributed to a greater increase in explosive strength of the lower body and, consequently, to a larger vertical jump height [62]. In this regard, Tornero-Aguilera and Clemente-Suárez [63] reported gains of ~3% in the horizontal jump after subjecting a group of soldiers to both resisted high-intensity interval training based on military exercises and to an endurance high-intensity interval training protocol based on running (>95% maximal HR), respectively (i.e., 3 series × 10 exercises × 30 s, with 30 s and 5 min rest). It is worth mentioning that these authors only found statistically significant increases in the lower limb explosive strength after performing the program based on resistance exercises.

The training program under study did not contribute to improving isometric hand-grip strength. The differences found before and after the training program were considered trivial (Table 3). These results differ from those obtained by Tornero-Aguilera and Clemente-Suárez [63], who showed improvements of 3.5–5% after implementing a high-intensity interval training program. It could be speculated that the characteristics of the exercises that made up their training favored the increase in hand-grip strength more than the one obtained in our study. However, it should be noted that the most remarkable improvements in their study were obtained using a training protocol that relied exclusively on high-intensity running. Therefore, it seems that the characteristics of the subjects in our research conditioned the results obtained. The WFFs routinely use different hand tools during their deployments [7] for extended periods [27], which contributes to their high hand-grip strength and can limit the improvements obtained with training programs [63]. In our work, the improvements found in the muscle resistance of the upper limb along with the trunk flexors and extensors (~13, ~34 and ~31%, respectively) were substantially lower (~135, ~75 and ~64%, respectively) than those reported in studies that followed similar training protocols with recreationally active women [64].

HICT not only led to improvements in the subjects’ strength but also improved WFFs’ ventilatory thresholds (Table 2, Figure 1). This finding was consistent with the aerobic performance changes previously informed after completing different high-intensity interval training interventions [65,66]. Robinson et al. [65] and Schaun et al. [66] described 2–13% increases in ventilatory thresholds after 12 and 48 high-intensity whole-body interval training sessions, respectively. The benefits of this type of interventions on aerobic performance measures could be a consequence of an improvement in muscle buffering capacity, body fat oxidation, increased mitochondrial density and an up-regulation of glycogen enzymes [31,65]. Despite this, our results did not show an increase in VO_2max_ after completing the training program. In contrast, the study by Schaun et al. [66] showed parallel improvements in RCT and VO_2max_ after performing 48 high-intensity interval training sessions. Although the RCT improved after 8 weeks of training (16 training sessions) in our study, this stimulus was not enough to induce improvements in VO_2max_. Subjects in the study by Schaun et al. [66] tripled the number of training sessions they underwent compared to those performed by the WFFs in our study. On the other hand, the high values of VO_2max_ analyzed at the beginning of the study in the WFFs (52.0 ± 8.2 mL kg^−1^ min^−1^) could condition its further enhancement [15]. Gist et al. [15] did not observe improvements in military VO_2max_ after applying a 4-week, high-intensity, total-body interval training program (12 training sessions). The initial values reported in these subjects were very similar (51.2 ± 5.6 mL·kg^−1^·min^−1^) to those in the present study. On the contrary, in firefighters who presented an initial VO_2max_ (40.8 ± 5.1 mL kg^−1^ min^−1^) lower than previously reported, increases of ~10% were obtained after 35 high-intensity training sessions [22]. The analyzed improvements in ventilatory thresholds could lead to greater work performance or work efficiency of WFFs. This fact could be especially relevant as the duration of their deployments increase [27], and in those situations where the WFFs have to adopt high and sustained work rates over time, such as the construction of fire lines, brush removal and clean-up activities. 

WFFs in this study improved their specific performance after the HICT intervention (Table 4). The Pack test time was reduced by ~12%. Similar improvements were reported in WFFs when their specific performance was assessed after 15 training sessions [26]. Similarly, Chizewski et al. [22] analyzed improvements after implementing a high-intensity training program in a six-event physical performance test to assess firefighters’ cardiorespiratory fitness and muscular resistance. The gains reported by these authors were ~20%, substantially higher than the one obtained in our study. This fact could be conditioned by the overall duration of the tests reported by Chizewski et al. [22], since they lasted 3–4 min. The improvements analyzed in each exercise ranged from 7.7–20.4%. The above results demonstrate the potential effect that training programs based on high-intensity training could have on firefighters’ productivity and work efficiency. Firefighters performing this type of training could double the chances of meeting the recommended physical employment standards compared to those who do not [23]. In addition, this type of training could potentially match the reality of the WFF experience, where work crews are made up of subjects of different ages and physical capabilities, and with an increasing number of women. Specifically, high-intensity training performed with exercises that use body weight as resistance can be considered an effective and safe alternative to improve physical condition and body composition in subjects of different ages and levels without the need to use a large amount of material [45,67].

The main limitation of this study was the lack of a control group with similar characteristics to those of WFFs who performed the intervention. More WFFs’ involvement was impossible due to the restrictions imposed by the recruitment agency. Despite this, initial physical fitness values between WFFs and WFF candidates were similar. On the other hand, the fact that WFF candidates attended a specific training camp could have conditioned some of the results obtained in this study. Nevertheless, the current findings show the potential benefit that HICT programs can have on WFFs’ physical fitness. Future research should compare the effectiveness of HICT against other types of exercise training. In the same way, involving female WFFs in these studies could help to determine if the sex of the subjects could condition the improvements analyzed in the HICT program. 

## 5. Conclusions

The results of this study show that high-intensity circuit training composed of specific work tasks is a good alternative to improve WFFs’ physical fitness and performance. Since the HICT protocol is more time efficient than the more conventional training models and does not require the use of extra material, it could be an excellent tool to improve the WFFs’ physical fitness during working hours at their bases, being able to combine it with their other functions and unpredictable time schedules. In addition, the HICT characteristics favor that the training sessions can be performed jointly by subjects of different physical fitness levels. This could facilitate the organization and design of training sessions at the WFFs’ bases and potentially favor the adherence of all subjects to the exercise programs.

## Figures and Tables

**Figure 1 ijerph-20-02073-f001:**
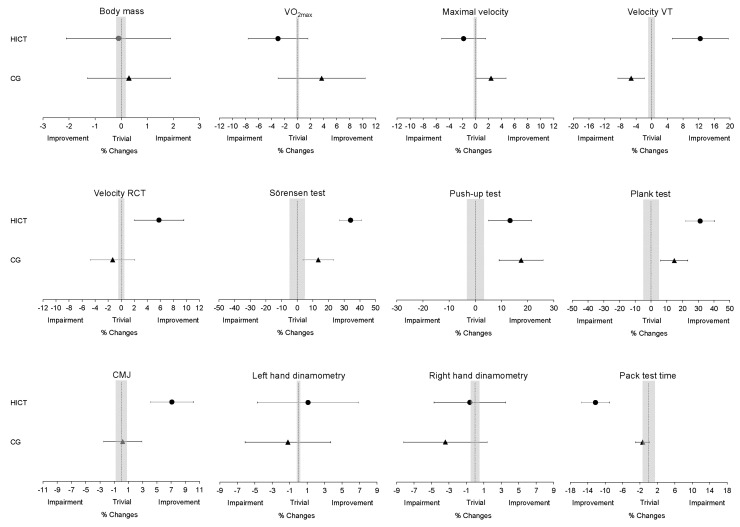
Relative changes for tests’ performance in the high-intensity circuit training group (HICT) and control group (CG). Values are mean ± 90% confidence interval. Trivial areas were computed from the meaningful changes.

**Table 1 ijerph-20-02073-t001:** High-intensity circuit training program.

Week	Work:Rest	Exercises (Tuesday)	Exercises (Thursday)
1	1:2 2 sets–8 exercises (20/40 s)/120 s	Burpees; Jumping jacks with fire swatter; Front plank touch shoulder; Hit with fire swatter (dominant side); CMJ; Hit with fire swatter (nondominant side); Mountain climbers; Skipping	Skipping; Push-up; Hit with fire swatter (dominant side); Lunge jump; Front plank; Burpees; Hit with fire swatter (nondominant side); Jumping jacks with fire swatter
2	1:1 2 sets–7 exercises (30/30 s)/120 s	Step up with Water backpack (20 kg); Hit with fire swatter (dominant side); Skipping; Thruster with Water backpack (20 kg); Hit with fire swatter (nondominant side)	Jumping jacks; Burpees; Hit with fire swatter (nondominant side); Skipping; Battle rope; Lunge jump; Hit with fire swatter (dominant side)
3	1:1 2 sets–8 exercises (30/30 s)/120 s	Jumping jacks; Kettlebell swing; Hit with fire swatter (nondominant side); Mountain climbers; Battle rope; Squat jump; Hit with fire swatter (dominant side); Skipping	Jumping jacks; Thruster with Water backpack (20 kg); Burpees; Skipping; Farmer Walk with Water backpack (20 kg); CMJ; Mountain climbers; Step up with Water backpack (20 kg)
4	2:1 2 sets–8 exercises (40/20 s)/120 s	Hit with fire swatter (dominant side); Thruster with Water backpack (20 kg); Burpees; Mountain climbers; Step up with Water backpack (20 kg); Front plank with push-up	Hit with fire swatter (nondominant side); Lunge jump; Hit with fire swatter (dominant side); Farmer Walk with Water backpack (20 kg); Jumping jacks; Thruster with Water backpack (20 kg)
5	2:1 2 sets–8 exercises (40/20 s)/120 s	Hit with fire swatter (dominant side); Jumping jacks with fire swatter; Hit with fire swatter (nondominant side); Skipping; Front plank touch shoulder; Hit with fire swatter (dominant side); Lunge jump; Battle rope	Hit with fire swatter (nondominant side); Thruster with Water backpack (20 kg); Skipping; Battle rope; Step up with Water backpack (20 kg); Mountain climbers; Hit with fire swatter (dominant side); Thruster with Water backpack (20 kg)
6	2:1 3 sets–6 exercises (40/20 s)/90 s	Hit with fire swatter (nondominant side); Farmer Walk with Water backpack (20 kg); Skipping; Hit with fire swatter (dominant side); Lunge jump; Thruster with Water backpack (20 kg)	Battle rope; Jumping jacks; Hit with fire swatter (nondominant side); Thruster with Water backpack (20 kg); Mountain climbers; Hit with fire swatter (dominant side)
7	2:1 3 sets–7 exercises (30/15 s)/90 s	Skipping; Burpees; Jumping jacks; Lunge jump; Thruster with Water backpack (20 kg); Mountain climbers; Push-up	Front plank touch shoulder; Squat jump; Hit with fire swatter (nondominant side); Lunge jump; Mountain climbers; Hit with fire swatter (dominant side); Jumping jacks
8	2:1 3 sets–7 exercises (30/15 s)/90 s	Hit with fire swatter (dominant side); Farmer Walk with Water backpack (20 kg); Skipping; Squat with Water backpack (20 kg); Mountain climbers; Hit with fire swatter (nondominant side); Burpees	Skipping; Hit with fire swatter (dominant side); Jumping jacks; Thruster with Water backpack (20 kg); Lunge jump; Hit with fire swatter (nondominant side); Battle Rope

**Table 2 ijerph-20-02073-t002:** Physiological characteristics of participants (mean ± *SD*).

	Group	Pre-Test	Post-Test	Change (%)	*p*-Value	Cohen’s *d*
VO_2max_ (ml·kg^−1·^min^−1^)	HICT	52.0 ± 8.2	50.8 ± 8.7	−3.0 ± 8.8	0.327	0.14 [trivial]
	CG	52.9 ± 6.3	55.4 ± 7.1	3.7 ± 12.2	0.334	0.37 [small]
Maximal HR (beats·min^−1^)	HICT	189 ± 11	185 ± 9	−1.7 ± 2.6	0.050	0.33 [small]
	CG	188 ± 12	191 ± 13	1.1 ± 2.8	0.258	0.35 [small]
Maximal velocity (km·h^−1^)	HICT	17.2 ± 2.7	16.8 ± 2.4	−1.8 ± 6.5	0.223	0.16 [trivial]
	CG	16.1 ± 1.4	16.5 ± 0.9	2.4 ±4.2	0.139	0.35 [small]
VO_2_ VT (ml·kg^−1·^min^−1^)	HICT	30.4 ± 6.7	34.3 ± 5.2	11.7 ± 20.8	0.108	0.65 [moderate]
	CG	33.7 ± 2.9	32.3 ± 5.6	−6.4 ± 16.9 *	0.367	0.31 [small]
%VO_2max_ VT	HICT	58.4 ± 9.5	68.4 ± 9.4	10.9 ± 15.0	0.054	1.06 [large]
	CG	64.1 ± 6.1	58.4 ± 7.4*	−5.7 ± 9.2 ***	0.099	0.84 [large]
HR VT (beats·min^−1^)	HICT	136 ± 20.0	145 ± 17.0	6.7 ± 12.6	0.178	0.46 [small]
	CG	147 ± 15	143 ± 19	−3.3 ± 4.7 *	0.077	0.23 [small]
% Maximal HR VT (%)	HICT	72.2 ± 9.8	77.9 ± 7.9	6.7 ± 9.0	0.072	0.64 [moderate]
	CG	77.9 ± 4.8	74.7 ± 5.8	−3.2 ± 3.4 **	0.022	0.60 [moderate]
Velocity VT (km·h^−1^)	HICT	9.3 ± 1.8	10.6 ± 1.5	12.4 ± 13.9	0.018	0.78 [moderate]
	CG	9.6 ± 0.9	9.1 ± 1.1	−5.2 ± 6.2 **	0.035	0.49 [small]
VO_2_ RCT (ml·kg^−1·^min^−1^)	HICT	41.8 ± 7.4	44.9 ± 6.9	7.0 ± 7.4	0.012	0.43 [small]
	CG	43.0 ± 5.0	44.0 ± 5.2	1.4 ± 10.4	0.526	0.19 [trivial]
%VO_2max_ RCT (%)	HICT	80.6 ± 9.9	88.9 ± 5.7	8.2 ± 11.1	0.034	1.02 [large]
	CG	81.8 ± 5.8	79.9 ± 9.2 *	−1.8 ± 7.8 **	0.505	0.87 [large]
HR RCT (beats·min^−1^)	HICT	169 ± 11	172 ± 11	1.9 ± 3.7	0.271	0.23 [small]
	CG	171 ± 12	169 ± 16	−1.5 ± 5.8	0.563	0.14 [trivial]
% Maximal HR RCT	HICT	89.8 ± 2.8	92.7 ± 2.5	3.3 ± 2.6	0.005	1.09 [large]
	CG	90.7 ± 2.2	88.6 ± 4.4 *	−2.2 ± 3.9 **	0.132	0.60 [moderate]
Velocity RCT (km·h^−1^)	HICT	13.5 ± 2.2	14.3 ± 1.9	5.7 ± 7.3	0.022	0.39 [small]
	CG	12.9 ± 0.6	12.8 ± 1.1	−1.3 ± 6.2 *	0.681	0.11 [trivial]

VO_2max_, maximum oxygen consumption; HR, heart rate; VT, ventilatory threshold; RCT, respiratory compensation threshold; %VO_2max_, percentage of VO_2max_ at which VT and RCT occur; HICT, high-intensity circuit training group; CG, control group. *, significant difference with HICT (*p* < 0.05). **, significant difference with HICT (*p* < 0.01). ***, significant difference with HICT (*p* < 0.001).

**Table 3 ijerph-20-02073-t003:** Results of anthropometric and muscle strength fitness tests (mean ± *SD*).

	Group	Pre-Test	Post-Test	Change (%)	*p*-Value	Cohen’s *d*
Body mass (kg)	HICT	76.7 ± 17.3	76.6 ± 16.9	−0.1 ± 2.9	0.954	0.00 [trivial]
	CG	76.4 ± 12.1	76.7 ± 12.2	0.3 ± 2.6	0.739	0.02 [trivial]
BMI (kg·m^−2^)	HICT	24.9 ± 4.7	24.8 ± 4.7	−0.2 ± 2.9	0.818	0.01 [trivial]
	CG	24.4 ± 2.6	24.5 ± 2.5	0.3 ± 2.5	0.718	0.03 [trivial]
Sörensen test (s)	HICT	75.1 ± 29.3	116.6 ± 48.7	34.1 ± 13.5	0.000	1.03 [large]
	CG	128.6 ± 46.4 *	147.3 ± 39.4	13.5 ± 17.8 **	0.064	0.43 [small]
Push-up test (rep)	HICT	36.2 ± 14.3	40.1 ± 12.6	13.3 ± 16.0	0.006	0.29 [small]
	CG	23.1 ± 6.5 *	38.3 ± 7.5 *	17.5 ± 15.3	0.006	2.10 [large]
Plank test (s)	HICT	126.8 ± 63.6	184.6 ± 69.6	31.2 ± 17.2	0.000	0.87 [large]
	CG	185.8 ± 69.4	216.6 ± 65.0	14.8 ± 15.7 **	0.014	0.45 [small]
CMJ (cm)	HICT	29.7 ± 6.2	31.8 ± 6.4	7.1 ± 5.8	0.007	0.33 [small]
	CG	33.3 ± 6.1	33.4 ± 6.3	0.2 ± 4.9 *	0.839	0.01 [trivial]
Dinam left (kg)	HICT	43.4 ± 9.5	43.9 ± 7.8	1.1 ± 11.2	0.650	0.06 [trivial]
	CG	45.1 ± 8.0	44.7 ± 7.3	−1.2 ± 8.9 *	0.737	0.05 [trivial]
Dinam right (kg)	HICT	47.0 ± 8.9	46.9 ± 7.2	−0.6 ± 7.9	0.901	0.01 [trivial]
	CG	48.2 ± 9.0	46.7 ± 7.9	−3.4 ± 8.8 *	0.319	0.02 [trivial]

BMI, body mass index; CMJ, countermovement jump; Dinam, manual dynamometry; HICT, high-intensity circuit training group; CG, control group. *, significant difference with HICT (*p* < 0.05). **, significant difference with HICT (*p* < 0.01).

**Table 4 ijerph-20-02073-t004:** Mean performance, physiological and perceptual responses during the Pack test (mean ± *SD*).

	Group	Pre-Test	Post-Test	Change (%)	*p*-Value	Cohen’s *d*
Completion time (min)	HICT	38.0 ± 3.4	33.5 ± 3.3	−12.2 ± 6.1	0.000	1.34 [large]
	CG	37.5 ± 3.0	37.1 ± 3.0 *	−1.4 ± 3.0 **	0.235	0.13 [trivial]
HR (beats·min^−1^)	HICT	147 ± 17	160 ± 12	7.9 ± 10.1	0.045	0.88 [large]
	CG	167 ± 18	189 ± 13	2.0 ± 4.6	0.127	1.40 [large]
% Maximal HR	HICT	79.4 ± 10.5	87.2 ± 3.9	7.8 ± 10.6	0.058	0.98 [large]
	CG	90.9 ± 7.2 *	89.4 ± 9.0	−1.5 ± 5.0 **	0.394	0.18 [trivial]
RPE	HICT	6.2 ± 0.4	8.0 ± 0.6	22.6 ± 7.1	0.000	3.53 [large]
	CG	7.3 ± 0.9 **	7.7 ± 0.5	4.6 ± 11.8 **	0.260	0.54 [moderate]

HR, heart rate; RPE, rating of perceived exertion; HICT, high-intensity circuit training group; CG, control group. *, significant difference with HICT (*p* < 0.05). **, significant difference with HICT (*p* < 0.01).

## Data Availability

The data presented in this study are available on request from the corresponding author. The data are not publicly available due to privacy restrictions.

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
