# Peer review of "Effect of a High-Intensity Circuit Training Program on the Physical Fitness of Wildland Firefighters"

_ijerph, 2023, doi:10.3390/ijerph20032073_

Round 1

Reviewer 1 Report

Dear authors.

In this manuscript you present an interesting intervention for a non-common population as are wildland firefighters. I consider that it will be very interesting to compare in future research the training method which is used in the intervention with others training method or unless with a control group. You remark this fact perfectly in the limitation of the study. However, I consider this manuscript presents an interesting proposal when it comes to training with forest firefighters, and it can be of interest for coaches who train with this population.

In my opinion, the manuscript is correct but I remark some questions and corrections that maybe you want to consider in order to improve the quality of the manuscript.

Abstract

-       Perhaps the physical capacities evaluated in the study (endurance, strength...) can be named so that it can be observed that a complete analysis of the condition has been carried out.

Introduction

-       Line 40-41: What kind of tests? Maybe a short description of the test that are commonly used for the selection of WFF can be add. It will help to the reader to know the physical requirements to perform this job. 

Methods

-       Can you add the number reference of the ethics commit?

-       I think that a better description of the physical activity of the participants can be added. Do they only train the two sessions of the training plan weekly? Or maybe do they train more than these two sessions by their own? What is the previous experience in strength training of these subjects? Maybe two sessions are good at the beginning to lead adaptations, but for people who train more days (4-5 days on a week) only two training can lead a reduction of performance.

-       You star with 30/30 work/rest ratio, and it evolve to 40/20. However, a decrease of 30/15 last two weeks was performed. What is the reason to reduce the volume this last two weeks? Please, describe this progression in the method section.

-       Line 168-171: do you use a metronome to mark the timing of the pushups? If you have not done, consider it for future research because the total time under tension can be limit the result of this test.

Results

-       Table 2: Please add “V” to O2max

-       Maybe can add * or bold for the statistically significant result. It is easier for the reader to detect the changes after the training period.

Discussion

-       Not improvement at VO2max but a improvement in VT was detected. You discuss perfectly that VO2max don’t improve in this period, but you can improve the discussion about the improvement in the VT. Maybe is for the time, total volume, and the rest of the intervals?

-       Please, add a short paragraph with practical applications of the study. 

Author Response

(Authors, AU) We would like to take this opportunity to thank reviewers for the time that they have taken to review our manuscript. Their valuable comments and rigorous feedback were very much appreciated, and they helped improve the manuscript quality. We hope that they will agree with changes made.

Review #1

- In this manuscript you present an interesting intervention for a non-common population as are wildland firefighters. I consider that it will be very interesting to compare in future research the training method which is used in the intervention with others training method or unless with a control group. You remark this fact perfectly in the limitation of the study. However, I consider this manuscript presents an interesting proposal when it comes to training with forest firefighters, and it can be of interest for coaches who train with this population.

(AU) We appreciate your comments. In this review we have substantially improved the original manuscript by introducing a control group. We have taken advantage of a recently conducted experimental phase with WFFs candidates with the same duration as the HICT intervention and using the same tests.

Abstract

- Perhaps the physical capacities evaluated in the study (endurance, strength...) can be named so that it can be observed that a complete analysis of the condition has been carried out.

(AU) we have introduce this in the abstract: “At the beginning and the end of the intervention, subjects' physical fitness (i.e., aerobic capacity, cardiorespiratory endurance, muscular resistance and explosive strength) was assessed through different tests.”

Introduction

- Line 40-41: What kind of tests? Maybe a short description of the test that are commonly used for the selection of WFF can be add. It will help to the reader to know the physical requirements to perform this job. 

(AU) we have added some sentences describing the test:

“Specifically in WFFs, to determine whether personnel are fit for duty, an increasing number of agencies (e.g., the USDA Forest Service; Australian Fire Agencies; British Columbia Forest Service in Canada or Ministry of Agriculture, Fisheries and Food in Spain) employ physical competency tests, such as the Pack test [13,14]. This test involves a 4.8 km hike over level terrain while carrying a 20.4 kg pack within 45 min. This test was designed to challenge an individual's muscular strength and cardiorespiratory fitness, mimicking the physiological strain encountered during wildland fire suppression using hand tools [13].”

Methods

- Can you add the number reference of the ethics commit?

(AU) It was added: “The test protocol was developed according to the Declaration of Helsinki guidelines for research on human subjects, and it was approved by the Ethics Committee of the University of León (025-2020, 22 July 2020).”

- I think that a better description of the physical activity of the participants can be added. Do they only train the two sessions of the training plan weekly? Or maybe do they train more than these two sessions by their own? What is the previous experience in strength training of these subjects? Maybe two sessions are good at the beginning to lead adaptations, but for people who train more days (4-5 days on a week) only two training can lead a reduction of performance.

(AU) This section was improved based on your suggestions. The changes made have been highlighted in the manuscript.

- You star with 30/30 work/rest ratio, and it evolve to 40/20. However, a decrease of 30/15 last two weeks was performed. What is the reason to reduce the volume this last two weeks? Please, describe this progression in the method section.

(AU) The work:rest ratio progression throughout the training program was based on previous recommendations (Machado et al. High-intensity interval training using whole-body exercises: training recommendations and methodological overview. Clin Physiol Funct Imaging. 2019;39(6):378–83). We have introduced a description of this progression in the methods section. The reduction in exercises volume during the last weeks was performed in parallel with an increase in the exercises intensity.

“The work:rest ratio was modified through the intervention period. It started with a ratio for beginners (1:2), from the second week the ratio was modified to a higher load pattern (1:1), and from the fourth week it was changed to a much more intense load ratio (2:1) [45].”

 - Line 168-171: do you use a metronome to mark the timing of the pushups? If you have not done, consider it for future research because the total time under tension can be limit the result of this test.

(AU) We did not use a metronome, but we gave the premise of not being able to stay in an isometric position for more than 1 second, the consequence of doing so being the end of the test. We appreciate your suggestion and it will be considered in future researches.

Results

- Table 2: Please add “V” to O2max

(AU) It was corrected.

- Maybe can add * or bold for the statistically significant result. It is easier for the reader to detect the changes after the training period.

(AU) We have indicated in bold type the significant differences between the pre- and post-tests. As a control group was introduced, the differences between the group that performed the intervention and the control were made through symbols.

Discussion

-  Not improvement at VO2max but a improvement in VT was detected. You discuss perfectly that VO2max don’t improve in this period, but you can improve the discussion about the improvement in the VT. Maybe is for the time, total volume, and the rest of the interval

(AU) This was discussed in the new version of the manuscript:

”HICT training not only led to improvements in the subjects' strength but also im-proved WFFs' ventilatory thresholds (Table 2, Figure 1). This finding was consistent with the aerobic performance changes previously informed after completing different high-intensity interval training interventions [65,66]. Robinson et al. [65] and Schaun et al. [66] described a 2-13% increases in ventilatory thresholds after 12 and 48 high-intensity whole-body interval training sessions, respectively. The benefits of this type of interven-tions on aerobic performance measures could be a consequence of an improvement in muscle buffering capacity, body fat oxydation, increased mitocondrial density and an up-regulation of glycogen enzymes [31,65].”

- Please, add a short paragraph with practical applications of the study. 

(AU) The practical applications were added.

Reviewer 2 Report

I am grateful for the opportunity to review this manuscript titled "Effect of a high-intensity circuit training program on the physical fitness of wildland firefighters”. The purpose of this study was to analyse the effects of high-intensity circuit training on the physical condition of a group of wildland firefighters. The data collected in this study may affirm or expand on available literature.

This study does not meet the quality standards to be considered for publication in IJERPH. It has been widely demonstrated that physical exercise improves sporting and professional performance, but what makes this study interesting? Furthermore, in my opinion, they make a serious design flaw by not having a control group.

Specific comments

1.     Authors should respect the style imposed by the journal (e.g., references, text indentation, ...).

Title and abstract

2.     The keywords 'high-intensity circuit training'; ‘wildland firefighters’; ‘training’ already appear in the title. Therefore, it would be convenient to make a modification in one of the two sections.

3.     It would be appropriate to specify the gender of the sample in the abstract.

4.     It would be appropriate for authors to introduce statistical values of the results in the abstract (i.e., p-value, effect size, mean and standard deviation, …). This can be compensated for by further summarizing the explanation of the experimental process.

Materials and Methods

5.     Were the same researchers who performed the assessments pre and post? In addition, it is not mentioned if the experimenters were blinded to the research question.

6.     Why didn't the authors include a control group?

Discussion

Although it is taken for granted when indicating that there is no control group, it would be appropriate to indicate that the experience of the WFFs in aerobic training, and the lack of experience in strength training, is a limiting aspect when extrapolating results to the population.

Author Response

(Authors, AU) We would like to take this opportunity to thank reviewers for the time that they have taken to review our manuscript. Their valuable comments and rigorous feedback were very much appreciated, and they helped improve the manuscript quality. We hope that they will agree with changes made.

Review #2

This study does not meet the quality standards to be considered for publication in IJERPH. It has been widely demonstrated that physical exercise improves sporting and professional performance, but what makes this study interesting? Furthermore, in my opinion, they make a serious design flaw by not having a control group.

(AU) In this review we have substantially improved the original manuscript by introducing a control group. We have taken advantage of a recently conducted experimental phase with WFFs candidates with the same duration as the HICT intervention and using the same tests. We have improved different sections of the manuscript and we hope that this new version has substantially improved your opinion.

Specific comments

  1. Authors should respect the style imposed by the journal (e.g., references, text indentation, ...).

(AU) This aspect was checked and corrected where necessary.

Title and abstract

  1. The keywords 'high-intensity circuit training'; ‘wildland firefighters’; ‘training’ already appear in the title. Therefore, it would be convenient to make a modification in one of the two sections.

(AU) This was modified: “Keywords: Performance; Exercise; Physical Activity; Strength Training; Endurance; Occupational Health”

  1. It would be appropriate to specify the gender of the sample in the abstract.

(AU) It was specified: “The study involved 9 WFFs (8 males and 1 female; 29.8 ± 2.8 years; 175.6 ± 6.7 cm) who completed a training program and 9 WFFs candidates (8 males and 1 female; 24.7 ± 6.6 yr, 176.5 ± 7.0 cm) as control group.”

  1. It would be appropriate for authors to introduce statistical values of the results in the abstract (i.e., p-value, effect size, mean and standard deviation, …). This can be compensated for by further summarizing the explanation of the experimental process.

(AU) We have taken your suggestion into account, the changes have been highlighted in the abstract.

Materials and Methods

  1. Were the same researchers who performed the assessments pre and post? In addition, it is not mentioned if the experimenters were blinded to the research question.

(AU) the researchers were not blinded to the research question and the same researchers performed the pre- and post-tests. This was indicated in the methods section.

  1. Why didn't the authors include a control group?

(AU) We only obtained permission to carry out the intervention with a WFF crew. Administrative restrictions imposed by the recruitment agency made it impossible to access more crews. In the current version of the manuscript this was overcome by using a group of WFF candidates as a control group.

Discussion

Although it is taken for granted when indicating that there is no control group, it would be appropriate to indicate that the experience of the WFFs in aerobic training, and the lack of experience in strength training, is a limiting aspect when extrapolating results to the population.

(AU) The WFFs who participated in the study had prior experience in both strength and aerobic training. Also, during the wildfire season they normally train at their bases, 3-4 days per week. This aspect was highlighted in the subject section.

Round 2

Reviewer 2 Report

I thank the authors for their efforts to improve the manuscript.